# MAUI (MBI Analysis User Interface)—An image processing pipeline for Multiplexed Mass Based Imaging

**Alex Baranski**[1], **Idan Milo**[2], **Shirley Greenbaum**[1], **John-Paul Oliveria**[1,3], **Dunja Mrdjen**[1], **Michael Angelo**[1]*, **Leeat Keren**[1,2]*

**1** School of Medicine, Department of Pathology, Stanford University, Stanford, California, United States of America, **2** Department of Molecular Cell Biology, Weizmann Institute of Science, Rehovot, Israel, **3** Department of Medicine, Division of Respirology, McMaster University, Hamilton, Ontario, Canada

\* mangelo0@stanford.edu (MA); leeat.keren@weizmann.ac.il (LK)

## Abstract

Mass Based Imaging (MBI) technologies such as Multiplexed Ion Beam Imaging by time of flight (MIBI-TOF) and Imaging Mass Cytometry (IMC) allow for the simultaneous measurement of the expression levels of 40 or more proteins in biological tissue, providing insight into cellular phenotypes and organization *in situ*. Imaging artifacts, resulting from the sample, assay or instrumentation complicate downstream analyses and require correction by domain experts. Here, we present MBI Analysis User Interface (MAUI), a series of graphical user interfaces that facilitate this data pre-processing, including the removal of channel crosstalk, noise and antibody aggregates. Our software streamlines these steps and accelerates processing by enabling real-time and interactive parameter tuning across multiple images.

## Author summary

High-dimensional Imaging technologies allow to simultaneously measure the expression levels of dozens of proteins in biological tissue, providing insight into single-cell phenotypes and organization *in situ*. Imaging artifacts, resulting from the sample, assay or instrumentation complicate downstream analyses and require correction by domain experts. Here, we present MAUI, a series of graphical user interfaces that facilitate this data pre-processing, including the removal of channel crosstalk, noise and antibody aggregates. MAUI accelerates and automates these steps, such that reproducible, high-quality data will be the input for subsequent stages of analysis.

This is a *PLOS Computational Biology* Software paper.

## Design and implementation

A major goal of biological research and medical diagnosis is to understand how cellular phenotypes at a single cell level relate to the multi-cellular structures in which they reside. This requires the ability to quantify the spatial distribution of multiple proteins or mRNA

**Data Availability Statement:** The data can be found at https://github.com/angelolab/MAUI.

**Funding:** IM is funded by the European Union's Horizon 2020 research and innovation programme

under the Marie Sk?odowska-Curie grant agreement No 890733. SG is supported by the Bill and Melinda Gates Foundation OPP1113682. JPO was supported by a Canadian Institutes of Health Research (CIHR) Postdoctoral Fellowship and a Banting Postdoctoral Fellowship. DM is supported by the Swiss National Science Foundation Early Postdoc Mobility Fellowship: P2ZHP3_181563 and Novartis Postdoctoral Fellowship: #18B067. LK is the Fred and Andrea Fallek President's Development Chair and is supported by the Azrieli Foundation, Enoch Foundation Research Fund, European Research Council Horizon 2020 starting grant 948811, Damon Runyon-Dale F. Frey Award 131248 and Israel Science Foundation personal grant 2481/20. This research was partially supported by the Israeli Council for Higher Education (CHE) via the Weizmann Data Science Research Center. MA was supported by the Department of Defence grant W81XWH2110143, the National Institute of Health grants 1-DP5-OD019822, 5R01CA22952903, 1U24CA22430901, 5U54CA20997105, 1R01AG056287, 1R01AG057915, 1U24CA224309, and the Bill and Melinda Gates Foundation. MA is a consultant and shareholder in IonPath Inc. The funders had no role in study design, data collection and analysis, decision to publish, or preparation of the manuscript.

**Competing interests:** I have read the journal's policy and the authors of this manuscript have the following competing interests: MA is a consultant and shareholder in IonPath.

transcripts across large regions of intact tissue. Recently, several new approaches have been introduced to allow this quantification. These can be broadly grouped to those that target mRNA, by either *in situ* hybridization [1,2] or sequencing [3–5] and those that target proteins, primarily using chromogen-, fluorescent- or mass-tagged antibodies [6–11]. These technologies vary in the number of measured parameters, size of images, speed of acquisition, resolution and compatibility with clinical specimens [12–14]. However, the data generated by these methods is ultimately similar, consisting of a set of registered images, one for each measured parameter. For all of these, imaging artifacts, resulting from the sample, assay or instrumentation complicate downstream analyses. While some properties of these artifacts are unique to a specific technology, others are common across several modalities. Here we present MAUI (MBI Analysis User Interface), a graphical user interface that wraps a series of image processing methods to eliminate imaging artifacts in Mass Based Imaging (MBI). Although MAUI was designed for MBI methods, such as Multiplexed Ion Beam Imaging by time of flight (MIBI-TOF) and Imaging Mass Cytometry (IMC), we also demonstrate its utility for fluorescent-based imaging.

We have previously described Multiplexed Ion Beam Imaging by time of flight (MIBI-TOF), a type of MBI method that uses secondary ionization mass spectrometry to image tissues stained with dozens of antibodies conjugated to heavy metal tags [15–17]. With MBI, tissue sections are stained with a cocktail of antibodies specific for the proteins of interest, each tagged with an isotopically-enriched elemental reporter. In MIBI-TOF, a primary ion beam scans across the sample to liberate the elemental reporters as secondary ions that are subsequently detected via orthogonal time of flight mass spectrometry. Ultimately, the result is a set of images, one for each antibody in the cocktail, where the intensity of signal in each pixel represents the number of ions (proportional to the amount of antibody and therefore target protein) detected in that physical location of the tissue (Fig 1).

While different factors can contribute to imaging artifacts, they can be broadly grouped into three classes. The first is the presence of channel crosstalk, whereby signal from one channel is detected at variable amounts in another channel. This can result from reporter impurities (e.g. Erbium 152 containing one percent of Erbium 151), peak tailing of high intensity reporters, polyatomic reporter adducts, or the presence of exposed gold slide [17]. The second is the presence of non-specific signal superimposed on top of the biological structures targeted by the antibody. This can typically be attributed to off-target antibody binding, high abundance organic adducts or instrumental noise. Depending on the intensity and spatial distribution of these sources, it includes signals that appear random, as well as dim patterns that correlate with histological features. The third is the occasional presence of aggregates, which show up as small bright specks of signal and are presumed to result from antibody aggregation. Notably, many of these issues are not specific to MIBI-TOF, and are encountered when using other highly multiplexed imaging modalities, including imaging mass cytometry (IMC) [6] or fluorescent approaches such as co-detection by indexing (CODEX) [9].

We have developed a series of algorithms that enables the removal of these imaging artifacts to facilitate downstream analysis [16]. The pre-processing proceeds in three stages, one for each of the types of artifacts described above: the removal of (1) channel crosstalk, (2) nonspecific staining, and (3) aggregates. Since the conditions under which these artifacts occur can vary between experiments, many parameters of the algorithms used are currently tuned by hand and evaluated by eye, leveraging the expert knowledge of pathologists and biologists. To facilitate this we present here MAUI, a graphical user interface that allows for real-time feedback between parameter adjustment and pre-processing output, drastically improving the data quality and efficiently reducing the time for further downstream analyses (Fig 2).

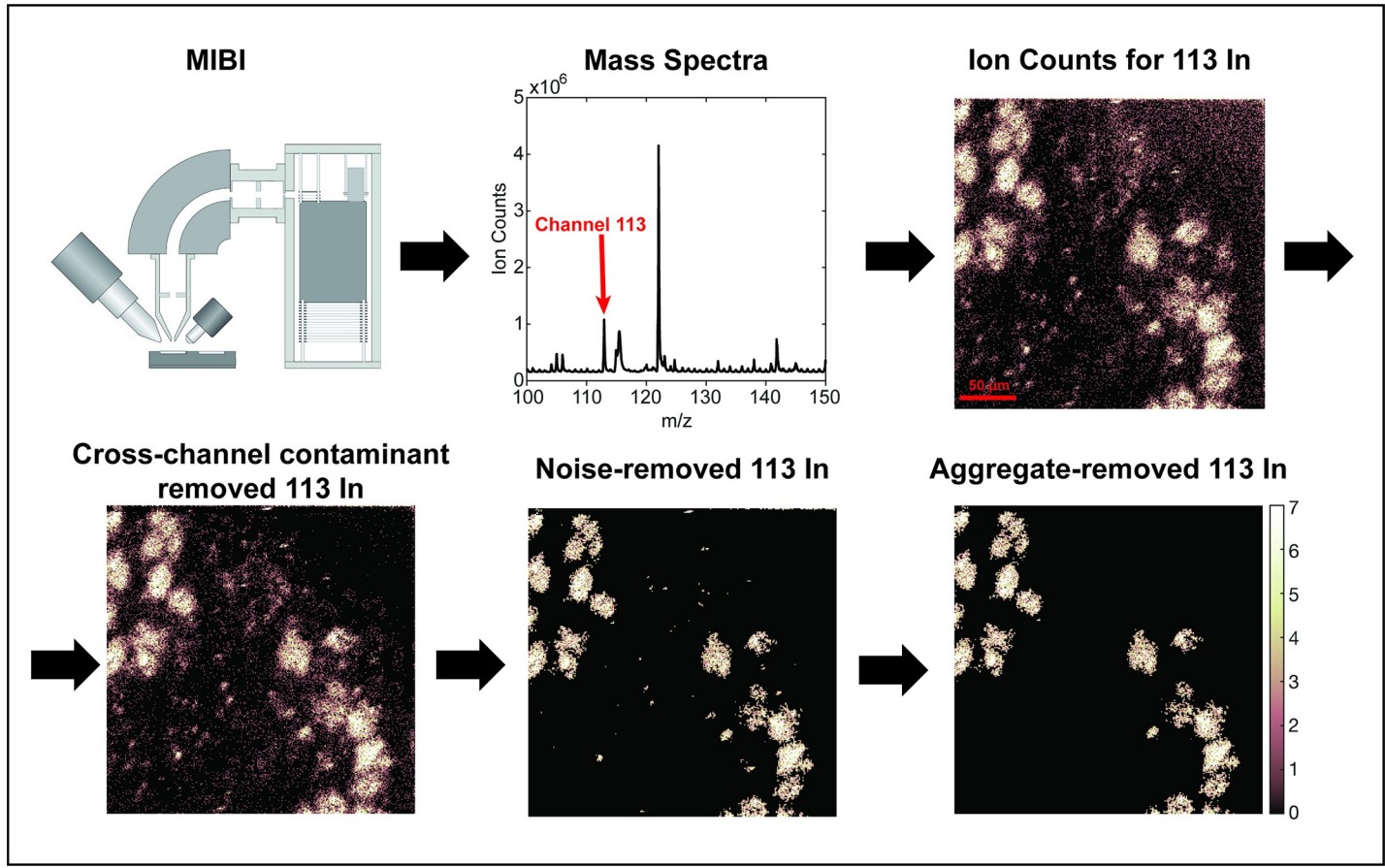

**Fig 1. Workflow of Multiplexed Ion Beam Imaging User Analysis Interface (MAUI).** Tissue specimens are stained using a mixture of antibodies labeled with elemental mass tags. The samples are rasterized by a primary ion beam that releases the metals of the bound antibodies as secondary ions, which are recorded by a Time of Flight Mass Spec. This results in an n-dimensional image depicting protein expression in the field. Image processing includes removal of cross-channel contamination, noise and aggregates.

## Results

We will demonstrate and discuss the functionalities of MAUI for MIBI data. Examples for similar analysis of IMC and CODEX are shown in S1 and S2 Figs respectively.

### Data structure

The input to the software is either a multipage TIFF or a folder of TIFF files. Each page in the multipage TIFF or each image in the folder, respectively, depicts the signal for a pre-specified mass range, which corresponds to a single conjugated antibody. Each processing step with MAUI will add an additional folder of TIFFs to the dataset, using as input the results of the previous processing step.

### Channel crosstalk

Channel crosstalk refers to scenarios where the signal of a source channel contaminates a target channel or is correlated with such contamination. Channel crosstalk in MBI can occur for a variety of reasons, some of which we will detail below. The first predominant cause for channel crosstalk is ionic contamination. In addition to the predominating monoatomic ion ($X^+$,

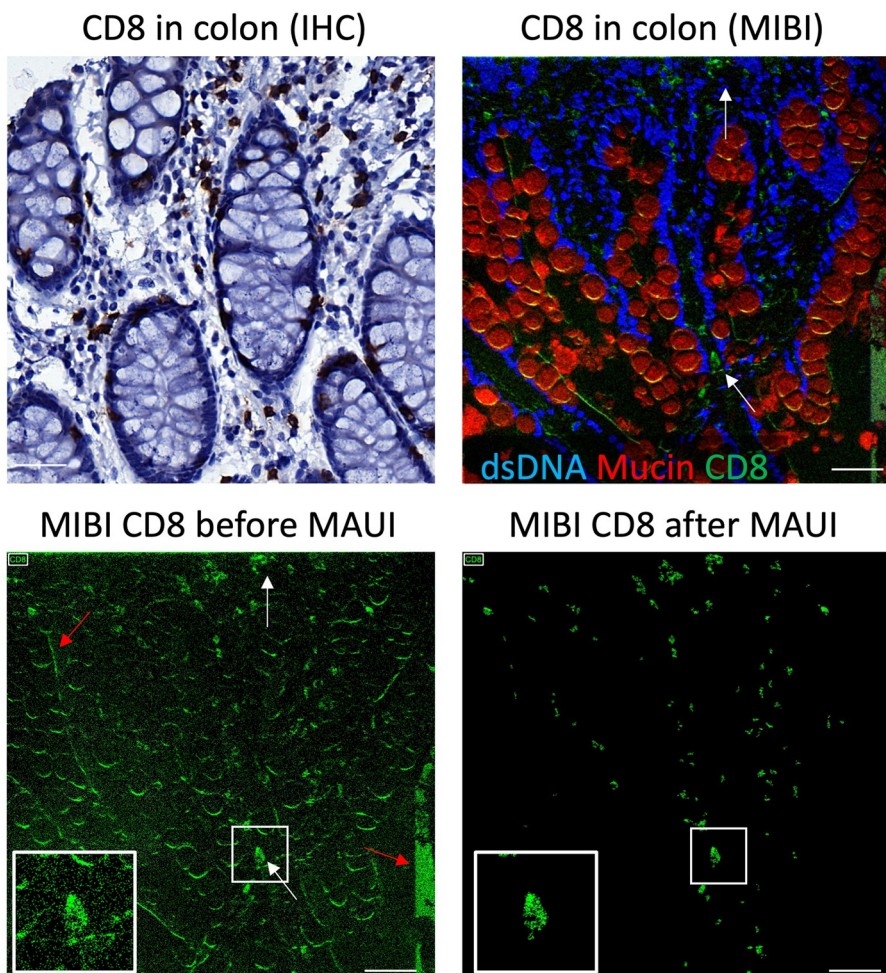

**Fig 2. Processing MIBI data using MAUI results in signal similar to Immunohistochemistry.** Shown are serial sections from human colon stained for CD8 by IHC (top left) or MIBI (top right). White arrows indicate cytotoxic T cells in the lamina propria of the gut. Red arrows indicate imaging artifacts including background and cross-channel contamination (bottom left). After processing by MAUI (bottom right) the CD8 signal mirrors the expected histological staining pattern by IHC. Scalebar equals 40μm.

where X = m/z), metal reporters used for antibody labeling can also produce relatively small amounts of metal hydrides ($XH^+$), oxides ($XO^+$), and hydroxides ($XOH^+$) [17] with masses of X+1, X+16, and X+17 respectively (Fig 3A). These polyatomic metal adducts can potentially contaminate heavier mass reporters. For example, 149Sm typically produces a metal oxide at 165 amu (149+16) with 4% the intensity of the monoatomic peak that can contaminate signal arising from an antibody tagged with 165Ho [17]. Alternatively, in early versions of the instrument we observed that bare regions of the slide often resulted in high-intensity signal across multiple mass channels. Since this signal is derived from the slide, it has a near-perfect overlap with components of the slide coating, e.g. gold (197Au), tantalum (181Ta) and silicon (28Si) channels (Fig 3B). The current generation of instruments does not suffer from this form of signal contamination (S3A Fig).

Channel crosstalk can also occur from isotopic impurities in the metal stocks used for conjugation. Many elements naturally exist as a mixture of isotopes, and while these have been enriched to high purity for conjugation, they still have varying amounts of contaminating

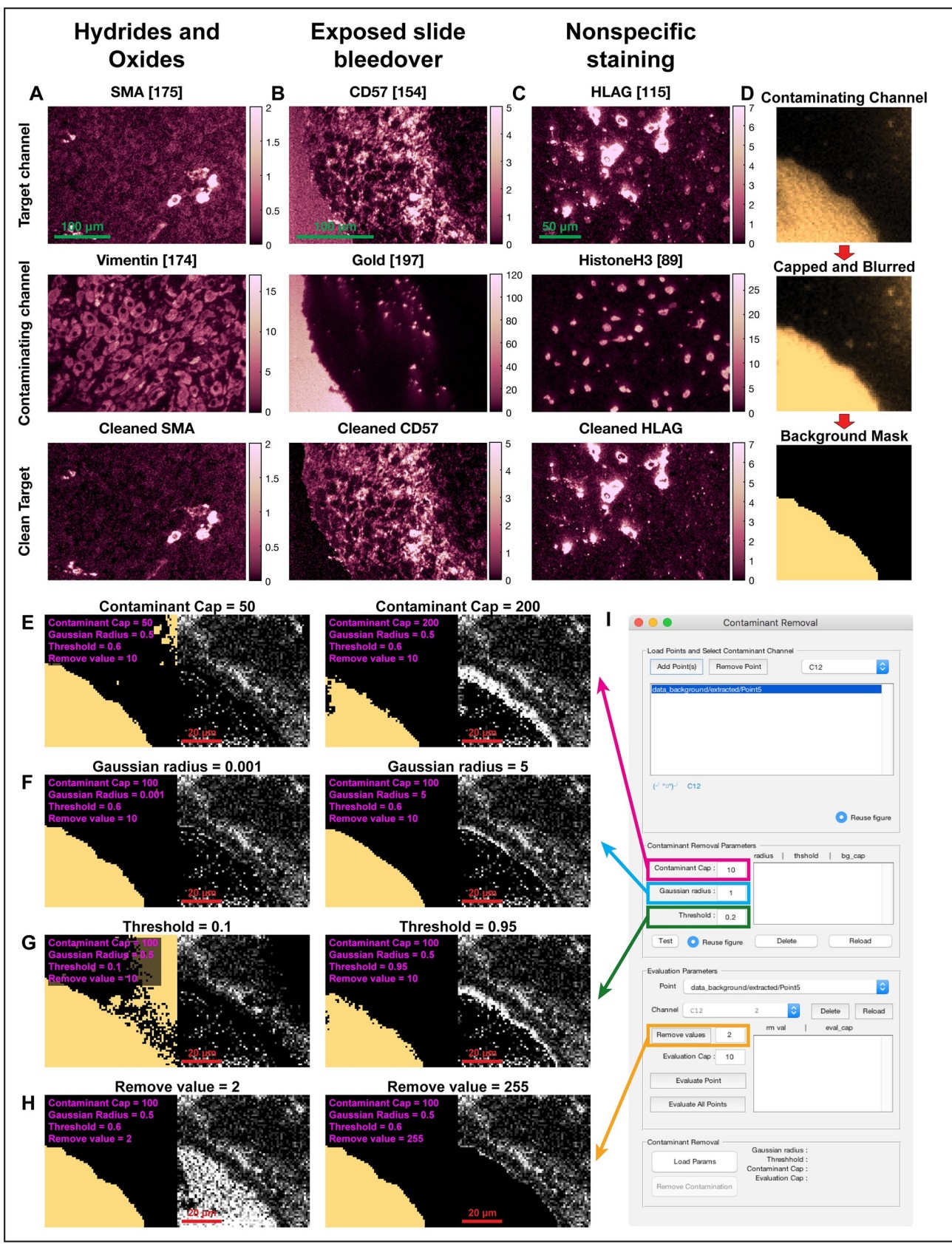

**Fig 3. Removal of cross-channel contamination. (A-C)** Shown are examples for different sources of cross-channel contamination. In all cases a target channel (top row) is contaminated by another channel (middle row) or is correlated with such contamination. MAUI allows to remove this contamination from the target image (bottom row). **(D)** The first step for removal of cross-channel contamination is to cap, blur and threshold the contaminating channel to generate a binary mask of the contamination. The second is to remove signal where the mask is positive. **(E-H)** Examples of the resulting image when varying the algorithm parameters, including the Contaminant Cap (E), Gaussian Radius (F), Threshold (G), and the Remove Value (H). **(I)** The GUI enables to rapidly evaluate these parameters across many points and channels.

isotopes of the same element. For example, the element samarium has 7 naturally occurring isotopes with an abundance ranging from 3.08% ($^{144}$Sm) to 26.74% ($^{152}$Sm). $^{152}$Sm can be purified to >98% by electromagnetic enrichment, but will still contain small amounts of the other samarium isotopes. Finally, channel crosstalk can result from systematic, off-target staining, resulting from the antibodies or metals used. This often presents as dim staining of biological structures that the antibody is not expected to bind to. For example, indium and aluminum preferentially localize to nuclei, and therefore these channels may present nonspecific nuclear staining (Figs 3C and S3B). In all these instances the contaminating signal in one channel mirrors real signal in another channel. Thus, one can use the latter to clean the signal in a target channel.

We designed MAUI to facilitate cleaning of a target channel using the signal of a contaminating source channel. The pipeline is divided into three parts, easily visualized in the user interface (UI). First, the contaminating channel is processed in several steps to generate a binary mask, whereby pixels that are positive for the signal will have a value of one, and pixels that are negative for the signal will have a value of zero. In this step, the user can tune several parameters to influence the resulting mask. Once users are satisfied with the mask, they can evaluate its performance in cleaning different target channels, a step that also involves user-defined parameters. These two steps can be iterated until the user is satisfied with the result. Finally, the user can apply the selected channels and parameters to remove the contamination in multiple images.

A detailed mathematical description of the contamination-removal algorithm is presented in the Methods section. Briefly, the first step of the algorithm is to generate a binary mask of the contaminating channel. Due to the sparse nature of mass spectrometry data, it is necessary to preprocess the contaminating channel before using it to clean the target channel. This includes capping the signal, smoothing it and binarizing it to generate a mask (Fig 3D). Each of these steps requires the user to input parameters that can be easily tuned and evaluated for their effect on the resulting mask. In the capping step, the user inputs a cap parameter, which sets the maximum value of the contaminating channel. Counts greater than the cap value will be set to the cap. This parameter allows for the homogenization of the signal intensity of the contaminating channel, by controlling overly bright signal (Fig 3E). The cap should be proportional to the expected intensity of the contaminating channel. Reducing its value will increase the number of pixels that are classified as positive for contaminant signal in the final mask.

Next, the capped contaminant signal is blurred using a radial Gaussian filter. The user inputs a *Gaussian Radius* parameter to set the standard deviation of the filter. This step allows for the homogenization and smoothing of the signal in neighboring pixels (Fig 3F). Increasing the radius increases the blurring and will most likely result in more positive pixels in the mask. We have found that a radius of one pixel was suitable in most cases.

Then, the capped and blurred image is normalized between zero and one. The *Threshold* parameter, which also takes values between zero and one, is then applied to generate a binary mask of the contaminating signal. Any pixel in the normalized contaminant channel above the threshold is set to 1, and any value below the threshold is set to 0. A higher threshold will reduce the number of positive pixels in the mask (Fig 3G). MAUI allows the user to view the original contaminant signal, while easily changing the algorithm parameters and evaluating the resulting mask. It also plots a histogram depicting the frequency of intensity values in the

normalized contaminant image to facilitate the user in picking a threshold that adequately captures the signal in the contaminating channel.

Once a binary mask of the contaminating channel has been generated, the user can now evaluate how this mask performs in cleaning any other channel using the *Evaluation Parameters* UI box. Cleaning is performed by subtracting a fixed value from all pixels in the target channel that were positive in the contaminant channel mask (Fig 3H). The value to remove is indicated by the user-defined parameter *Remove Value*. After removal, any value in the target channel that is below zero is set to zero. The user is given the option to remove a specific value, rather than, for example, zeroing out these pixels, because sometimes these pixels also have real signal superimposed with the contaminant. It is possible to zero out these pixels by choosing an arbitrarily large *Remove Val*. MAUI gives the user the option to evaluate all the parameters in cleaning different target channels and in different images.

Finally, once the users are satisfied with the chosen channels and parameters, they can use them to remove contamination in multiple images (Fig 3I). MAUI will create a new folder and populate it with the contamination-removed versions of the loaded data. These can be used as input for further preprocessing, including additional rounds of contamination removal. The software will also save a log file documenting the parameters used, to allow reproducibility of the analysis.

## Noise removal

Image noise is traditionally defined as unwanted variation in image information, which deteriorates image quality. In antibody-based visualization of tissues it can result from various factors, ranging from instrumentation to tissue quality to nonspecific binding of antibodies, which generate weak signal that may follow histological patterns. For our purposes, we define noise as any signal outside of the biological structures the stain is expected to mark, and that does not specifically overlap with any of the other channels in the panel. Therefore, it is not possible to remove it by the contamination-removal technique described above.

Here, we apply a K-Nearest-Neighbor (KNN) approach to noise filtering [18]. The approach takes advantage of the relatively low density of noise counts as compared to signal counts (high signal to noise ratio). It is based on the assumption that the presence of the intended biological target at a location will increase the odds of the antibody binding at that location, and that higher ion counts in and near a pixel indicate a higher probability of the associated biological signal being present at that location. We find this approach particularly suitable for MIBI data due to the sparse nature of the data.

The denoising algorithm consists of two stages: first, the count density near each pixel is estimated using an intensity-normalized version of KNN, and second, the signal from low-density pixels is filtered. KNN estimates density by calculating the distances from a specific pixel $x_{(i,j)}$ in the image to the closest $k$ non-zero pixels. The shorter these distances are, the higher the density in that location. The intensity-normalized version performs similarly, but takes into account the count value in each positive pixel. Thus, instead of measuring the distances to the $k$ closest positive pixels, it will calculate the distance to the $k$ closest individual ion counts, whereby several counts can be found in the same pixel.

For example, in Fig 4A we demonstrate the calculation of the 5-nearest-neighbors of the central pixel of the image, with a value if $x_{(i,j)} = 2$. We think about this value as constituting two separate events. Thus, the first neighbor is inside of the same pixel, with a distance of zero. The next nearest neighbor is one pixel to the right, with a distance of one. The next three nearest-neighbors are two pixels away above, each of these with a distance of two (Fig 4B).

We define the ADK as the average distance to the k nearest neighbors. The average distance to the 5-nearest-neighbors in the example above is *(0+1+2+2+2)/5 = 1.4*. For a given value of

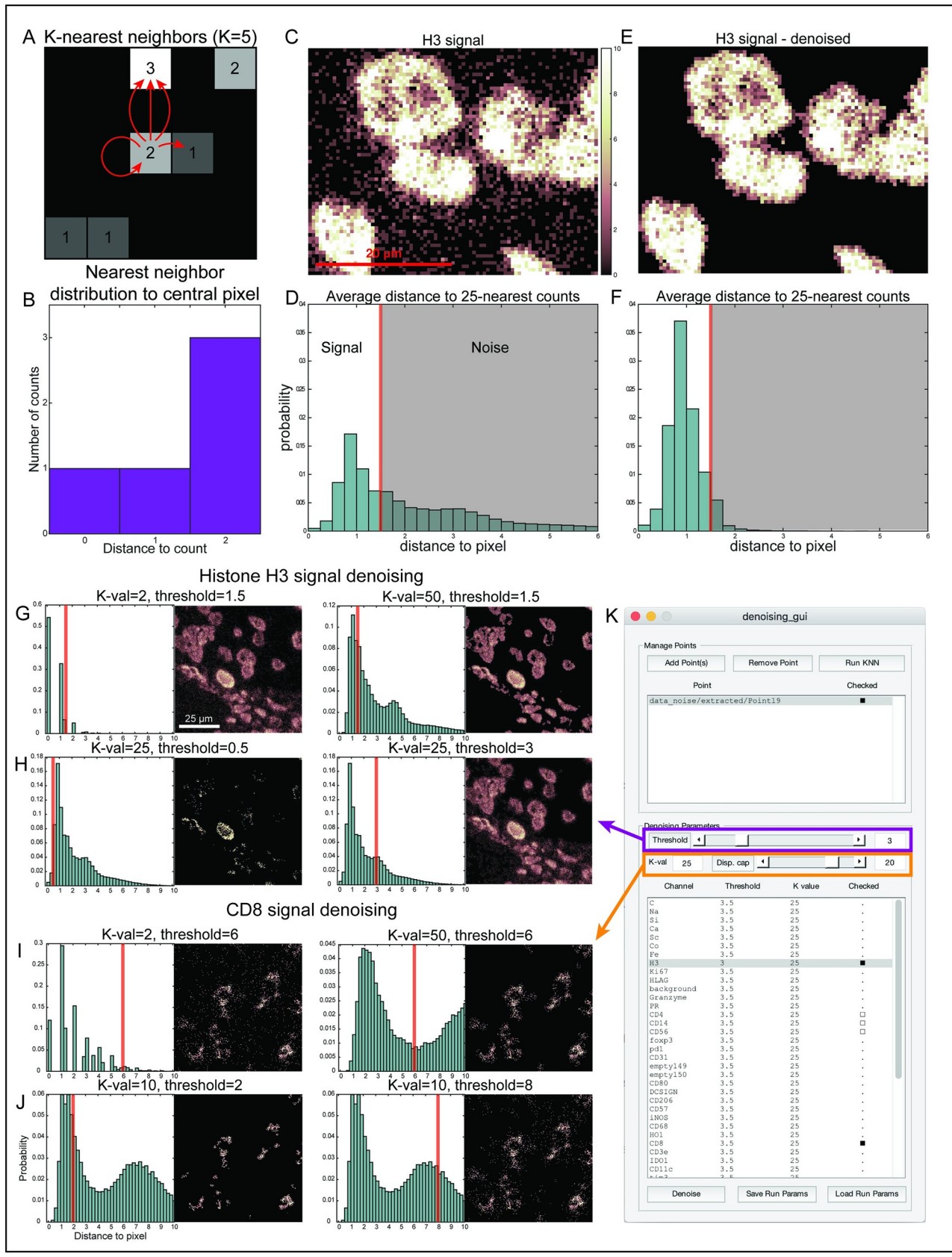

**Fig 4. Removal of noise using a K-Nearest-Neighbor (KNN) algorithm.** **(A-B)** Signal density at each pixel is determined using its K nearest neighbor counts. Shown is an example of the 5 nearest counts (A) and their distances (B) to the central pixel. **(C)** An example of staining of Histone H3 with noise surrounding the nuclear signal. **(D)** Shown is the frequency of the average distance to the 25 nearest counts for all pixels in (C). Signal is characterized by short distances (high density), whereas noise is characterized by long distances (low density). The red line is the chosen threshold for denoising, and pixels with large distances to their neighbors will be set to zero. **(E)** Histone H3 signal after denoising. **(F)** The average KNN-distribution for Histone H3, same as in (D), after denoising. **(G-H)** Examples of the resulting image of Histone H3 when varying the algorithm parameters, including the K-value (G) and Threshold (H). **(I-J)** Examples of the resulting image of a membranous protein, CD8, when varying the algorithm parameters, including the K-value (I) and Threshold (J). **(K)** The GUI enables to rapidly evaluate these parameters across many points and channels.

k, we then calculate the ADK for every pixel $x_{(i,j)}$ in the image. Fig 4C depicts an image of Histone H3 (HH3), while Fig 4D depicts the corresponding distribution of ADK values for all positive pixels in the image, using a *k* value of 25. This is a bimodal distribution in which low ADK values correspond to high density of counts and are therefore attributable to signal, whereas high ADK values correspond to low density of counts, and therefore are considered as noise. The actual denoising involves picking a threshold used to separate "noise" from "signal". Pixels with higher count densities (and therefore lower ADK values) are considered to be true signal and are left unchanged, while pixels with lower count densities (and therefore higher ADK values) are considered to be noise and their count value is set to zero. For example, using a threshold of 1.5 on the HH3 image shown in Fig 4C, any pixel with an ADK above 1.5 is set to zero, resulting in the image and corresponding distribution of ADK values shown in Fig 4E and 4F, correspondingly. A full mathematical derivation of the process is detailed in the Methods section.

MAUI is designed to facilitate, streamline and expedite the process of denoising using the intensity-normalized KNN approach for multiple images at a time (Fig 4G–4K). The *k*-value and threshold are user-defined parameters, which can be easily adjusted for each marker at the discretion of the domain-expert. MAUI generates real-time feedback as the *k*-value and threshold are adjusted, making it fast and easy to intuit which value ranges are most effective for denoising a particular dataset. When a given marker is selected on a given image, the ADK distribution for the *k*-value is plotted, along with an indication of the current threshold value. Easy scrolling between loaded images allows rapid evaluation of these properties on different images. The raw data is plotted, along with the denoised data and the difference between the two. Because the dynamic range of MIBI images can be quite skewed (significant biological information can be contained within ion-count differences of 1 or 2 counts, while outliers of 10 or 20 counts are possible), a display cap parameter is included for easy visual display of the data. We find that both the ADK histogram and the images are useful tools in guiding users' choice of parameters.

Notably, current experience with this approach has demonstrated that the algorithm is quite robust to the specific value of *k* and for the most part does not need to be changed given the same imaging conditions. On the other hand, the threshold is dependent on the distributions of signal and noise for each target and is usually defined for each channel separately. Once appropriate *k* and threshold values have been selected for each channel, the entire dataset can be easily denoised using these parameters. MAUI will create a new folder and populate it with the noise-removed versions of the loaded data. These can be used as input for further processing. The software will also save a log file documenting the parameters used, to increase reproducibility.

## Aggregate removal

Aggregates are concentrated areas of high counts that have been reported in both MIBI and IMC data [16,19]. These spots are uncorrelated with any biological structures and are thought

to result from conglomerations of antibodies. Aggregates can interfere with downstream analysis, particularly of weak channels, where they may be falsely interpreted as positive staining (Fig 5A). A more detailed discussion of aggregate identification is presented in the Methods section.

The procedure for removing aggregates takes advantage of their small size, much smaller than real signal of biological structures, and involves finding small continuous areas of positive counts. First, the image is blurred using a Gaussian filter, and then the blurred image is thresholded using a cutoff of zero to create a binary mask. The blurring has the effect of merging patches of signal that are near each other, usually indicative of them being part of a larger structure, such as a cell or vessel. Then, once a mask has been generated, all connected objects in the binary mask and their sizes are identified. Any object below a size threshold selected by the user is considered to be an aggregate, and the values of the counts in its constituent pixels are set to zero (Fig 5A).

MAUI allows the user to pick for each channel in the data the radius of the Gaussian blur as well as the size threshold for objects to be considered aggregates, listing the chosen parameters for each channel in a list (Fig 5B and 5C). MAUI will graphically display in real time the raw data and the aggregate-removed data, resulting from the currently chosen parameters. Empirically, we find that mild blurring (e.g. radius = 1) and a single threshold are often adequate across channels and datasets. Once an appropriate set of parameters has been chosen, MAUI allows to easily apply these parameters to all images in the dataset. Just as with the contamination removal and noise removal steps, MAUI will create a new folder and populate it with the aggregate-removed versions of the loaded data, as well as a log file recording the parameters.

## Availability and future directions

High-dimensional imaging technologies are poised to revolutionize tissue histology. These new modalities need specialized computational pipelines to facilitate pre-processing such that clean, high-quality data will be the input for subsequent stages of analysis. Here, we surveyed some of the common issues associated with data collected with multiplexed immunohistochemistry imaging platforms. We have described the simple computational solutions developed in our lab to address these issues and make MBI data suitable for subsequent analysis. MAUI, the graphical user interface that wraps these solutions, has substantially improved the speed and ease in which biologists and clinicians in our lab interact with the data and perform these labor and computationally-intensive steps in a reproducible and efficient manner. MAUI can be downloaded from https://github.com/angelolab/MAUI. We expect future developments in assays, instrumentation and computation to further enhance the quality of multiplexed-imaging data, making it more accessible to the large scientific community.

## Methods

### Contaminant removal

For a given channel of MBI data $f(x,y)$ (1) the user finds an associated contaminating channel $h(x,y)$ (2). The user then picks a cap value $c$ which is used to create a capped version of the contaminating channel $h_{capped}(x,y)$ (3). The user then picks a standard deviation $\sigma$ for a Gaussian blur $g(x,y)$ (4), which is convolved with $h_{capped}$ to create a blurred version $h_{blurred}$ (5). This image is then rescaled to have a maximum of 1, producing an image $h_{rescaled}$ (6). The user then picks a threshold value $t$ which is used to binarize $h_{rescaled}$, producing a binary mask that can be used for contaminant removal, $h_{mask}$ (7). Finally, the user selects a removal value $v$. At any pixel location $(x,y)$ where $h_{mask}(x,y)$ is 1, we remove $v$ from $f(x,y)$, producing $f_{removed}$ (8). Anywhere $f_{removed}<0$ is set to zero, since negative values of counts are nonsensical. This produced

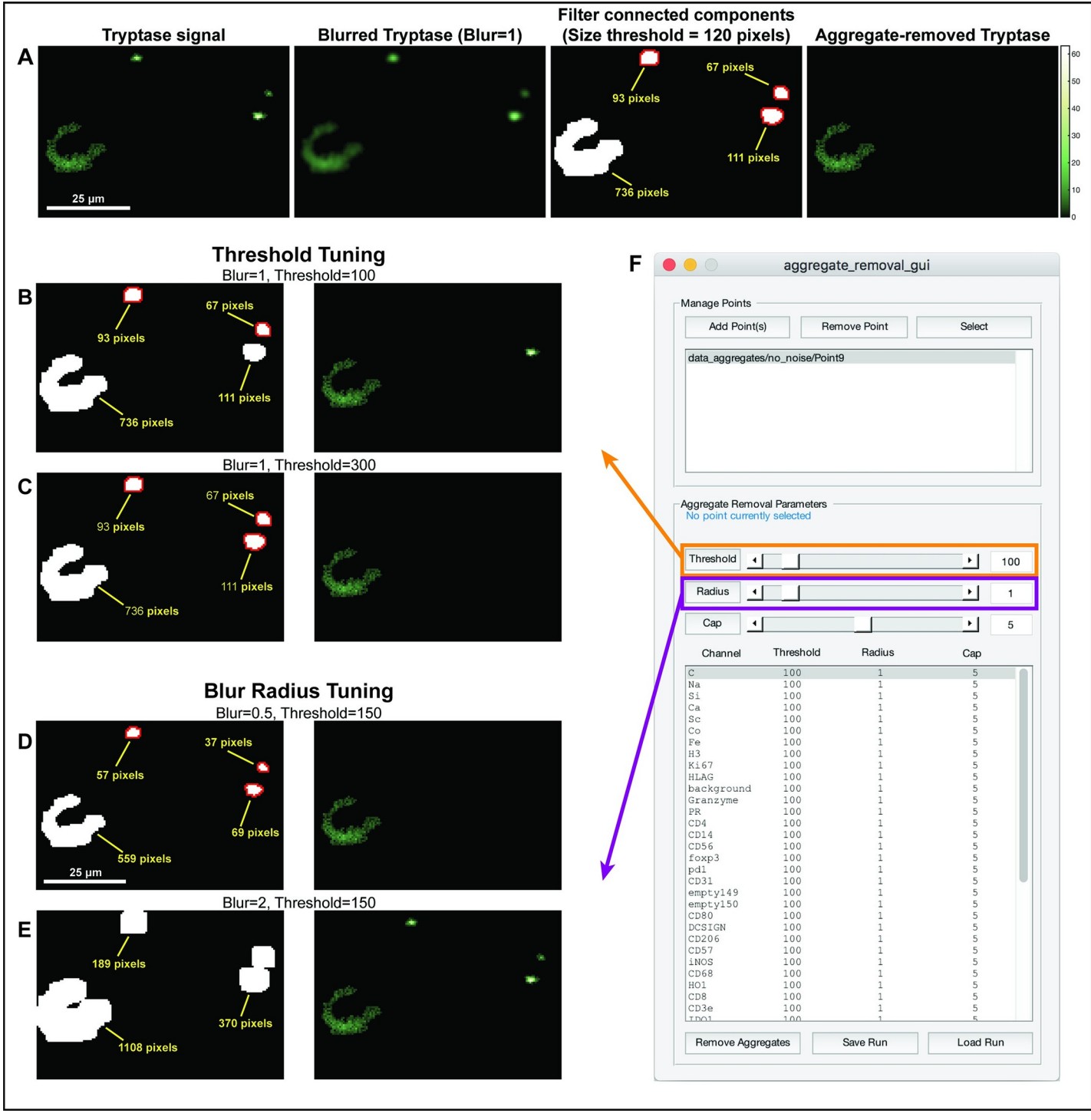

**Fig 5. Removal of aggregates. (A)** An overview of the aggregate-removal process. Left: An image of denoised Tryptase signal. Aggregates are evident in the top right corner. The image is blurred with a Gaussian filter and then binarized. Objects are extracted from the binary image, and any object small enough is considered an aggregate and removed. **(B-C)** Examples of the resulting image of Tryptase when varying the algorithm parameters, including the Threshold (B,C) and Blur radius (D-E). **(F)** The GUI enables to rapidly evaluate these parameters across many points and channels.

the final contaminant-removed MBI data, $f_{clean}$ (9).

$$f(x, y) = Target\ MBI\ data \tag{1}$$

$$h(x, y) = Contaminating\ MBI\ data \tag{2}$$

$$h_{capped}(x, y) = \begin{cases} h(x, y), h(x, y) \leq c \\ c, h(x, y) > c \end{cases} \tag{3}$$

$$g(x, y) = \frac{1}{2\pi\sigma^2} e^{-\frac{x^2+y^2}{2\sigma^2}} \tag{4}$$

$$h_{blurred} = g * h_{capped} \tag{5}$$

$$h_{rescaled} = \frac{h_{blurred}}{\max_{(x,y)}[h_{blurred}(x, y)]} \tag{6}$$

$$h_{mask}(x, y) = \begin{cases} 1, h_{blurred}(x, y) \geq t \\ 0, h_{blurred}(x, y) < t \end{cases} \tag{7}$$

$$f_{removed}(x, y) = \begin{cases} f(x, y), h_{mask}(x, y) = 0 \\ f(x, y) - v, h_{mask}(x, y) = 1 \end{cases} \tag{8}$$

$$f_{clean}(x, y) = \begin{cases} f_{removed}(x, y), f_{removed}(x, y) \geq 0 \\ 0, f_{removed}(x, y) < 0 \end{cases} \tag{9}$$

## Noise removal

We let $f(x,y)$ be the function that maps pixels in an image to ion counts on a particular channel of MBI data (10). We define a function $C(x,y)$ which takes in a single pixel $(x,y)$ and outputs a list with $f(x,y)$ copies of $f(x,y)$ (11). We let this function be defined on lists of pixels $(a,b,c,\ldots)$ such that the output is the concatenation of the outputs of the function on each of the individual pixels in the input list (12). We then define another function $N(x,y)$ which outputs a list of all pixels in the image ordered by their Euclidean distance (denoted by $\|\cdot\|$) from $(x,y)$ (13). We then let $\mathcal{N}(x, y) = C(N(x, y))$ (14) and take only the first $K$ entries, defining $\mathcal{N}_K(x, y)$, which we call the $K$-nearest neighbors of $(x,y)$ (15). We then calculate the average distance to the $K$-nearest neighbors of each pixel $(x,y)$ denoted by $A_K(x,y)$ (16). To clean the data, the user picks a threshold $t_n$, which is used to separate "noise" counts from "signal" counts. Any pixel with an average distance to its $K$-nearest neighbors more than the threshold is treated as noise and zeroed-out, otherwise its value is left unchanged (17).

$$f(x, y) = MBI\ data, x \in [1, w], y \in [1, h] \tag{10}$$

$$C((x, y)) = ((x, y), (x, y), \ldots (x, y))\ \text{s.t.}\ |C((x, y))| = f(x, y) \tag{11}$$

$$C((a, b, c, \ldots)) = C(a) \oplus C(b) \oplus C(c) \oplus \ldots \tag{12}$$

$$N(x, y) = ((x, y)_1, (x, y)_2, \ldots (x, y)_{w \times h}) \text{ s.t. } \forall (x, y)_i \in N(x, y), \tag{13}$$

$$\| (x, y)_{i+1} - (x, y) \| > \| (x, y)_i - (x, y) \|$$

$$\mathcal{N}(x, y) = C(N(x, y)) \tag{14}$$

$$\mathcal{N}_K(x, y) = ((x, y)_i | i \in [1, K]) \tag{15}$$

$$A_K(x, y) = \frac{1}{K} \sum_{i=1}^{K} \| (x, y)_i - (x, y) \| \tag{16}$$

$$f_{clean}(x, y) = \begin{cases} f(x, y), A\_K(x, y) \leq t_n \\ 0, A\_K(x, y) > t_n \end{cases} \tag{17}$$

## Aggregate removal

We let $f(x,y)$ be MBI data, as in (10). The user picks a standard deviation $\sigma$ for a Gaussian filter $g$ (18), which is convolved with our data to produce $f_{blurred}$ (19). We then binarize this image to create $f_{mask}$ (20).

We use a built-in function in MATLAB called *regionprops*, here denoted as $f_{object}$, which takes in a binary mask and outputs all connected sets of pixels with a value of 1 (thus any set of pixels in the input with a value of 1 which are connected to each other will be considered a single object). All such objects are returned as outputs of the function (21). The user defines a threshold $t_a$ for the size of aggregate objects, where objects with fewer pixels than $t_a$ are counted as aggregates (22). Finally, all pixels in the data the user is cleaning that are part of an aggregate object are zeroed-out (23).

$$g(x, y) = \frac{1}{2\pi\sigma^2} e^{-\frac{x^2+y^2}{2\sigma^2}} \tag{18}$$

$$f_{blurred} = g * f \tag{19}$$

$$f_{mask}(x, y) = \begin{cases} 0, f_{blurred}(x, y) \leq 0 \\ 1, f_{blurred}(x, y) > 0 \end{cases} \tag{20}$$

$$f_{object}(M) = \{o_1, o_2, \ldots o_n\} \text{ s.t. } o_i \text{ is an object in } M \tag{21}$$

$$f_{aggregate}(M) = \{o \in f_{object}(M) | |o| < t_a\} \tag{22}$$

$$f_{clean}(x, y) = \begin{cases} f(x, y), \nexists o \in f_{aggregate}(M) \text{ s.t.} (x, y) \in o \\ 0, \exists o \in f_{aggregate}(M) \text{ s.t.} (x, y) \in o \end{cases} \tag{23}$$

Judging whether large *clumps of signal* is true signal vs aggregation can range from easy to non-trivial depending on the antibody, tissue and aggregation state. Currently, this process depends to a large extent on experts with domain knowledge. We commonly employ the

 

following guidelines to decide if a signal is real or an aggregate: (A) *Familiarity with the expected staining pattern of the antibody*. Whereas for some antibodies we expect small, punctate staining, which is difficult to differentiate from aggregates (e.g. Granzyme B), most of the antibodies that we use stain larger cellular structures. We use knowledge on the expected staining patterns, both intra-cellularly (e.g. nuclear vs. membrane), cellularly (e.g. expected coexpression of markers) and histologically (e.g. expected staining in some anatomical locations and not others) to decide whether a "clump of signal" is real or not. While there may be ambiguous cases, for the most part it is quite clear. (B) *Comparison between different tissues*. In our experience, once an antibody vial begins to aggregate, aggregates will be observed in all tissues stained with that vial. As such, if one suspects that signal is actually an aggregate, it is useful to see whether aggregates are present in this channel across additional images. (C) *Design of specific experiments to check for aggregates*. When one is uncertain whether an antibody is aggregated, it is possible to design specific staining experiments to address this. For example, staining a negative control tissue that is not expected to express the target protein. Alternatively, one can stain serial sections and see whether the signal repeats between sections.

## Supporting information

**S1 Fig. Analysis of IMC data using MAUI.** The top left panel shows an image of CD68 from an IMC dataset (Jackson et al., Nature 2020). Staining intensity is shown as a heatmap from blue (low) to high (yellow). White arrows denote real CD68 staining, as validated by coexpression of CD45. Colored arrows denote various imaging artifacts including cross-channel contamination (red), noise (green) and aggregates (orange). Each row in the image shows one stage of processing by MAUI, including removal of cross channel contamination (top), denoising (middle) and removal of aggregates (bottom) as detailed. The final image is shown in the bottom right. Scalebar equals 30μm.
(JPEG)

**S2 Fig. Analysis of CODEX data using MAUI.** The top left panel shows an image of CD8 from a CODEX dataset (Schurch et al., Cell 2020). Staining intensity is shown as a heatmap from blue (low) to high (yellow). White arrows denote real CD8 staining, as validated by coexpression of CD3 and CD45. Colored arrows denote various imaging artifacts including cross-channel contamination (red), noise (green) and aggregates (orange). Each row in the image shows one stage of processing by MAUI, including removal of cross channel contamination (top), denoising (middle) and removal of aggregates (bottom) as detailed. The final image is shown in the bottom right. Scalebar equals 50μm.
(JPEG)

**S3 Fig. Underlying reasons for cross channel contamination. (A)** Shown is an image for the mass range of 128–132, which has no labeled signal in it (left column) for the first (top) and second (bottom) generations of the instrument. In the first-generation instruments this background signal (right column) mirrors the bare slide and therefore the gold channel (middle column). In the second-generation instruments (bottom), this slide-specific background is mitigated and the background only has salt-and-pepper noise. **(B)** MIBI-TOF staining of human FFPE tonsil with free $^{115}$Indium and antibodies for Histone H3 and CD45 shows nuclear localization of $^{115}$Indium. Scalebar equals 10μm.
(JPEG)

## Author Contributions

**Conceptualization:** Michael Angelo, Leeat Keren.

 

**Data curation:** Alex Baranski, Idan Milo, Shirley Greenbaum, John-Paul Oliveria, Dunja Mrdjen.

**Formal analysis:** Alex Baranski, Leeat Keren.

**Funding acquisition:** Michael Angelo.

**Methodology:** Alex Baranski, Michael Angelo, Leeat Keren.

**Project administration:** Michael Angelo, Leeat Keren.

**Resources:** Alex Baranski, Idan Milo, Shirley Greenbaum, John-Paul Oliveria, Dunja Mrdjen, Michael Angelo, Leeat Keren.

**Software:** Alex Baranski.

**Supervision:** Michael Angelo, Leeat Keren.

**Validation:** Alex Baranski, Idan Milo, Shirley Greenbaum, John-Paul Oliveria, Dunja Mrdjen, Michael Angelo, Leeat Keren.

**Visualization:** Alex Baranski, Leeat Keren.

**Writing – original draft:** Alex Baranski, Michael Angelo, Leeat Keren.

**Writing – review & editing:** Idan Milo, Michael Angelo, Leeat Keren.

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
