## [Decision Letter · Decision Letter 0]

21 Dec 2020

Dear Dr. Keren,

Thank you very much for submitting your manuscript "MAUI (MBI Analysis User Interface) - An image processing pipeline for Multiplexed Mass Based Imaging" for consideration at PLOS Computational Biology. As with all papers reviewed by the journal, your manuscript was reviewed by members of the editorial board and by several independent reviewers. The reviewers appreciated the attention to an important topic. Based on the reviews, we are likely to accept this manuscript for publication, providing that you modify the manuscript according to the review recommendations.

Sincerely,

Dina Schneidman

Software Editor

PLOS Computational Biology

[LINK]

Reviewer's Responses to Questions

**Comments to the Authors:**

Reviewer #1: Baranski and co-workers here present an interactive graphical user interface adaptation of the image processing methods originally described in Keren et al, 2018 Cell. Specifically, the software, Mass Based Imaging Analysis User Interface (MAUI), streamlines image processing from Multiplexed Ion Beam Imaging (MIBI) developed by the Angelo lab.

In addition to the MAUI software, the important work presented here also details and shows convincing examples of the results of each step performed in preliminary MIBI image processing. The authors indicate that the work can potentially extrapolate to other forms of mass spectrometry imaging, and is certainly appropriate for the scope of the journal, given the momentum in adopting various multiplexed imaging platforms in the scientific community (including the methods related to SIMS and the IMC) and the lack of alternative software that enables image cleanup without the use of scripting languages.

I have a number of minor suggestions and comments that I hope the authors can address, in addition to showcasing broad utility of MAUI.

1. In Results 2.2, the authors state tantalum as 182Ta. It would appear that 181Ta is the stable isotope of tantalum.

2. Will the authors be able to clarify how channel 89Y would cause contaminating signals in channel 115In? It does not appear to fall into the typical +1, +16 and +17 contaminants.

3. It is unclear whether MAUI would allow multiple rounds of sequential contamination removal (eg removing a -1 and then a -16 on the same channel). If so, would the authors be able to elaborate further in the text?

4. In Results 2.3, the authors state that the KNN approach here “has been shown to have superior performance to other filters”. It would be more convincing to have some representative images for other methods of denoising, including DBSCAN, another commonly used density-based clustering method.

5. It is unclear how aggregation removal would be judged by a user. IE how would one decide whether large “clumps” of signal is true signal vs aggregation? It would also be useful to elaborate how the signal density of aggregates will be taken into consideration for the clean up.

Additionally (and optionally), it would certainly strengthen the manuscript to see examples of how MAUI processing of publicly available IMC images or other mass-spec imaging data (such as ToF-SIMS, nano-SIMS or OrbiSIMS).

Thank you for your work in putting together a very nice piece of software for the community.

Reviewer #2: The authors developed an MBI Analysis User Interface (MAUI) that facilitates the processing of image data, including the removal of channel crosstalk, noise, and antibody aggregates, and provides a software resource for efficient user data processing of data acquired by Multiplexed Ion Beam Imaging by time of flight (MIBI-TOF).

Recent developments and advancements in MIBI-TOF and Imaging Mass Cytometry (IMC) have provided innovative multiplex molecular imaging methods in the life sciences and have attracted the interest of many scientists in this field. There is no doubt that authors' contributions are important, especially for advanced lifescience researchers who are trying to adopt them.

However, after careful perusal of this manuscript, I believe that several issues need to be clarified. My main comments are as follows.

1) There is no doubt that high-dimensional imaging techniques using mass spectrometry are becoming increasingly important for histopathology, with several different instruments and methods available, including MIBI and IMC. However, since there are many non-specialist readers (especially researchers in the life sciences) who are interested in this field, I would like to see more detailed description of the instruments and methods that are currently available for users to choose from. Moreover, the instruments and methods that can be covered by MAUI should be clearly described.

2) The reviewer is interested in whether a comparison with fluorescence microscopy images using fluorescent secondary antibodies would be useful in validating images acquired with MIBI-TOF and processed with MAUI.

In particular, it was felt that comparison with fluorescence images could clearly show whether the inherent problems in mass spectrometry caused by channel crosstalk are solved by this software. On the other hand, the problems of non-specific signals and antibody aggregation are considered to be common problems in fluorescent immunostaining methods, and it should be discussed whether it is reasonable to solve these problems with this software in MIBI-TOF images.

**Have all data underlying the figures and results presented in the manuscript been provided?**

Reviewer #1: Yes

Reviewer #2: Yes

PLOS authors have the option to publish the peer review history of their article (what does this mean?). If published, this will include your full peer review and any attached files.

Reviewer #1: **Yes: **Sizun Jiang

Reviewer #2: No
---

## [Decision Letter · Decision Letter 1]

17 Mar 2021

Dear Dr. Keren,

We are pleased to inform you that your manuscript 'MAUI (MBI Analysis User Interface) - An image processing pipeline for Multiplexed Mass Based Imaging' has been provisionally accepted for publication in PLOS Computational Biology.

Best regards,

Dina Schneidman

Software Editor

PLOS Computational Biology

Reviewer's Responses to Questions

**Comments to the Authors:**

Reviewer #1: Dear Authors,

Thank you for the very nicely written revision comments and edits. These revisions full address my concerns and I recommend publication without reservations.

I have a minor comment on a typo in Figure S2 (Vimenting -> Vimentin) which may not be addressed in subsequent copy editing as it shows up as an image and not text.

Congratulations on a very robust study!

Reviewer #2: My comments have been addressed and the MS is improved. I have no additional comments.

**Have all data underlying the figures and results presented in the manuscript been provided?**

Reviewer #1: Yes

Reviewer #2: None

PLOS authors have the option to publish the peer review history of their article (what does this mean?). If published, this will include your full peer review and any attached files.

Reviewer #1: **Yes: **Sizun Jiang

Reviewer #2: No

---

## [Editor Report · Acceptance letter]

13 Apr 2021

PCOMPBIOL-D-20-01128R1 

MAUI (MBI Analysis User Interface) - An image processing pipeline for Multiplexed Mass Based Imaging

Dear Dr Keren,

I am pleased to inform you that your manuscript has been formally accepted for publication in PLOS Computational Biology. Your manuscript is now with our production department and you will be notified of the publication date in due course.

With kind regards,

Andrea Szabo
